# Mucoadhesive Electrospun Fibre-Based Technologies for Oral Medicine

**DOI:** 10.3390/pharmaceutics12060504

**Published:** 2020-06-02

**Authors:** Jake G. Edmans, Katharina H. Clitherow, Craig Murdoch, Paul V. Hatton, Sebastian G. Spain, Helen E. Colley

**Affiliations:** 1School of Clinical Dentistry, 19 Claremont Crescent, University of Sheffield, Sheffield S10 2TA, UK; jedmans1@sheffield.ac.uk (J.G.E.); k.clitherow@btinternet.com (K.H.C.); paul.hatton@sheffield.ac.uk (P.V.H.); h.colley@sheffield.ac.uk (H.E.C.); 2Department of Chemistry, Brook Hill, University of Sheffield, Sheffield S3 7HF, UK; s.g.spain@sheffield.ac.uk

**Keywords:** bioadhesion, mucosa, local therapy, electrospinning, oral cavity, drug delivery, inflammation, infections, pain relief

## Abstract

Oral disease greatly affects quality of life, as the mouth is required for a wide range of activities including speech, food and liquid consumption. Treatment of oral disease is greatly limited by the dose forms that are currently available, which suffer from short contact times, poor site specificity, and sensitivity to mechanical stimulation. Mucoadhesive devices prepared using electrospinning offer the potential to address these challenges by allowing unidirectional site-specific drug delivery through intimate contact with the mucosa and with high surface areas to facilitate drug release. This review will discuss the range of electrospun mucoadhesive devices that have recently been reported to address oral inflammatory diseases, pain relief, and infections, as well as new treatments that are likely to be enabled by this technology in the future.

## 1. Introduction

The oral cavity plays a vital role in day-to-day life, including speech, mastication, eating, drinking as well as other sensory functions. These functions are all underpinned by healthy oral tissues—the impairment of which due to disease can vastly reduce quality of life [1,2]. Treatment of oral diseases can be difficult, where frequent high topical doses applied across the whole of the oral cavity, or drugs delivered systemically, are the main treatment options for often, small affected areas of the oral mucosa. Some localised topical treatment methods using gels or creams are currently on the market; however, many of these only have a transient therapeutic effect due to limited drug retention on the affected mucosa.

The field of oral medicine faces several challenges in finding an appropriate drug delivery system that offers sustained drug release to directly target the disease site or lesion, not least because the moist environment in the mouth and flexibility of the mucosal tissue surfaces makes adhesion difficult. This represents a major unmet clinical need as there are currently no effective commercially available drug delivery systems that fully address these problems. Electrospinning thin fibrous patches for local oral drug delivery may be ideal to overcome these challenges, where the manufactured fibres have a high surface area to simultaneously allow increased drug release and mucoadhesive interaction with the tissue. In recent decades, numerous scientific studies have shown that electrospun systems can act as drug delivery vehicles [3,4]. More recently, electrospun oromucosal drug delivery systems are being developed for local treatment of oral diseases. This review introduces the requirements, manufacture, and characterisation of electrospun mucoadhesive systems suitable for application to the oral mucosa and discusses materials currently in development for use in oral medicine. Research so far has almost exclusively been aimed at providing pain relief or treating infections and inflammatory diseases. This review also discusses how electrospun mucoadhesives could be further developed for these applications and identifies potential future treatments that could be enabled by this technology.

## 2. Oral Mucosa

### 2.1. Structure

The oral mucosa is the mucous membrane lining the oral cavity and consists of a stratified squamous epithelium, basement membrane, lamina propria and submucosa (Figure 1) [5]. The epithelium typically consists of five layers (stratum basale, stratum spinosum, stratum granulosum and stratum corneum) depending on level of keratinisation. The epithelium is made up of oral keratinocytes that originate in the stratum basale (basal layer), where they divide by uneven mitosis. In this process, one daughter cell remains in the basal layer and is attached to the basement membrane, where it can undergo further rounds of cell division. The other daughter cell migrates apically into the stratum spinosum; more commonly termed the spinous layer. Once in the spinous layer, the keratinocytes lose the ability to divide and begin a programme of terminal differentiation as they progress into the stratum granulosum. Here, the keratinocytes contain membrane coating granules that extrude lipids and these, along with the low intracellular volume, act as a highly efficient permeability barrier against hydrophilic materials [6]. The keratinocytes finally enter the stratum corneum, where they can either shed their nuclei and increase keratin production to become keratinised or retain their nuclei and become non-keratinised, before eventually being lost to the oral cavity by desquamation. The hard palate, dorsum of the tongue and the gingiva are covered in masticatory, keratinised stratified squamous epithelium whilst the inner lips (labial mucosa), cheek (buccal) mucosa, soft palate and floor of the mouth are covered in lining mucosa that consists or a non-keratinised stratified squamous epithelium. Although oral keratinocytes make up 95% of the total cell number in the oral epithelium, other important cells are present including dendritic cells that perform important immunosurveillance roles and sensory Merkel cells. The surface of the epithelium is bathed in mucins (highly glycosylated proteins) and inorganic salts are primarily secreted by sublingual salivary glands. These cause the gelation of the outer layer into a protective and lubricating layer of mucous with a thickness of 40–300 µm, followed by an additional 70 µm coating of saliva [5]. 

Basolateral to the oral epithelium is the lamina propria, a fibrous connective tissue layer where oral fibroblasts produce elastin and type I and III collagen fibres to form the extracellular matrix. The lamina propria also contains blood vessels, glands and nerves. The submucosa beneath the lamina propria, which may or may not be present depending on the region of the oral cavity, consists of loose connective tissue that connects the oral mucosa to the underlying muscles. Healthy oral lining mucosa, such as that of the buccal mucosa, consists of approximately 40 to 50 cell layers [7], with an average thickness of 294 ± 68 μm [8]. However, epithelial thickness varies at different mucosal sites, with the floor of the mouth having the thinnest mucosa. 

Diseased oral tissue usually originates in the epithelium. For example, malignancy occurs due to genetic defects in the basal keratinocytes leading to uncontrolled cell division. Auto-inflammatory diseases such as oral lichen planus occur as a result of immune cell-mediated damage of the stratum basale, whilst fungi such as *Candida albicans* can infect the upper epithelial layers causing oral candidiasis or denture stomatitis. Therefore, the epithelium is the main drug delivery target for the treatment of most oromucosal diseases [9].

### 2.2. Permeation

The permeability of the oral epithelium is dependent on its thickness, lipid content in the granular layer and degree of keratinisation. In general, the higher lipid content in keratinised regions lowers the permeability [7]. Oral mucosal permeability is lower than in the intestine due to increased thickness and reduced surface area of the epithelium. There are multiple routes for a drug to pass through the oral mucosa and the predominant route depends on the physicochemical properties of the drug [10]. Small-molecule lipophilic drugs such as fentanyl [11] often partition into cell membranes, and so diffuse predominantly through the epithelial cells (transcellular route) and often cross the oral mucosa efficiently without any permeation enhancers. In the case of ionisable small-molecule drugs, such as lamotrigine, the pH of the delivery system may be adjusted to favour the non-ionised form to promote transcellular diffusion [12]. Larger and more hydrophilic compounds, including peptides, tend to favour transport around keratinocytes (paracellular route) and are usually less well absorbed [10]. For certain drugs, transcellular transport across the oral mucosa may occur via carrier-mediated transport. For example, there is evidence that monocarboxylate [13] and glucose [14] transporters are expressed on the keratinocyte cell surface; therefore, drugs that are substrates for these transporters may have increased epithelial uptake.

### 2.3. Current Oromucosal Drug Delivery Systems

A variety of commercially available formulation types target the oral cavity and these have been reviewed in detail by Hearnden et al. [15]. Mouthwashes are commonly used for the local delivery of antimicrobials [16]. Mucoadhesive gels, pastes, and hydrogel-forming films are also mostly used for local topical delivery or to form protective layers over wounds, for example to treat ulcers and sores [17]. Gels have also been trialled for the systemic delivery of analgesics [18] and anti-hypertensives [19]. Buccal tablets and lozenges are used for both topical and systemic delivery and may include mucoadhesives. Here, drugs are released as the tablet dissolves, offering exposure times of up to 30 min [20]. Buccal tablets have been used for several drugs including opioid painkillers [20], nitroglycerin, and steroid hormones for hormone replacement therapy [15]. These require the hormone to permeate through the oral mucosa. Buccal tablets have also been used for the local delivery of antifungals to treat oral candidiasis [21].

These existing dose forms offer relatively short exposure times and tend to deliver the drug non-specifically across the whole oral cavity. Mucoadhesive gels and tablets offer improved retention over rinses but are prone to becoming dislodged by mechanical stimulation and are likely to interfere with speech. Drug doses tend to be inconsistent due to variations in saliva flow and swallowing [20]. The oral mucosa is a highly challenging site for the development of a mucoadhesive dose form due to constant saliva flow and mechanical forces. There is a clear need for new formulations that allow specific delivery of a well-defined drug dose to the oral mucosa. Electrospun materials are an interesting emerging technology for this application, due to their flexibility and thinness in comparison to tablets, which is expected to result in improved comfort and retention. Their high surface area and porosity allows for rapid swelling enabling controlled drug release and an increased number of mucoadhesive interactions with the mucosa.

## 3. Electrospun Mucoadhesive Materials

### 3.1. Electrospinning

Electrospinning uses a high voltage (5–30 kV) to produce polymer fibres, with diameters ranging from two nanometres up to several micrometres from a polymer solution or melt [22]. So far, at least 12 electrospun medical devices are in late stages of regulatory or market approval, with the majority being used as surgical grafts or for tissue regeneration [23,24,25,26,27,28,29,30,31,32,33,34]. The technique is particularly promising for drug delivery because of its versatility and the high surface area of the resulting nanofibre mesh, which allows an active compound to be incorporated and released at a controlled rate by either diffusion or degradation of the nanofibres [3]. A typical electrospinning set-up consists of a spinneret needle loaded with polymer solution, a high voltage power supply, and a grounded collector plate (Figure 2). The power supply injects charge into the solution causing a stream to accelerate away from the tip due to the electrical repulsion exceeding surface tension. The point of eruption is called the Taylor cone. A syringe pump drives the syringe at a controlled flow rate to keep the spinneret tip filled. Polymer entanglements increase viscosity leading to the formation of a continuous fibre rather than droplets. The polymer stream drawn away from the needle tip undergoes whipping motions caused by electrostatic repulsions within the stream [35]. The solvent evaporates rapidly during the flight, leaving a mat of polymer nanofibres on the collector plate (Figure 3).

Several different types of collector can be used, most commonly a static plate or a rotating drum. Rotating drums allow a more uniform membrane thickness or the collection of aligned fibres, which are typically less porous and have different release and mechanical properties [22,36]. Templates may also be used to produce membranes with a three-dimensional patterned surface [37]. Solution and processing parameters affect fibre morphology, including diameter and the incidence of defects. These effects are reviewed in detail elsewhere [22]. In general, higher solution conductivity and lower solution viscosity are associated with narrower fibres while viscosity must be suitable to counteract solution surface tension effects and allow a continuous stream to flow.

Modifications to the electrospinning technique include the production of fibres with multiple polymer domains using coaxial, emulsion, or side-by-side electrospinning (Table 1) [38,39,40]. For drug delivery, these can be used to improve the processability of a drug-containing phase using a second polymer phase or to influence release rate or adhesion strength by encapsulating the drug within a sheath. Recently, there has been interest in high-throughput nanofibre production to allow more economical mass production. Needleless electrospinning involves injecting charge into a trough or surface containing polymer solution, causing many polymer jets to be produced simultaneously [41,42]. Centrifugal electrospinning involves a heated rotating cylinder as the spinneret, which ejects molten polymer through narrow outlets under a combination of electrostatic and centrifugal force. This potentially enables high-throughput solvent-free production but is challenging to optimise [43]. Currently, arrays of uniaxial needles are often used to increase scale in industrial settings.

### 3.2. Biocompatible Polymers

It is important that the device and its components be non-irritant and non-toxic both in the oral cavity and in the gastrointestinal tract, in case it is accidentally swallowed. Both natural and synthetic polymers can be electrospun into biocompatible drug delivery membranes. The most commonly used synthetic polymers are biodegradable polyesters such as poly(lactic acid) (PLA), poly(glycolic acid) (PGA), poly(lactic-*co*-glycolic acid) (PLGA), and polycaprolactone (PCL) [52]. These polymers, which have been extensively studied for use in orthopaedic devices where biodegradability is often desirable [53], are typically soluble in organic and halogenated solvents such as chloroform, dichloromethane (DCM), dimethylformamide (DMF), and tetrahydrofuran [54] and have high tensile strengths. Biodegradable polyesters are also suitable for oromucosal devices, as they are often non-inflammatory over the relevant timescales, easily processed, easily sterilised, and have good shelf lives. These polymers are not typically adhesive but can be blended with a mucoadhesive or used as part of a composite system to improve residence time. Poly(ethylene glycol) (PEG), poly(vinyl alcohol) (PVA) and poly(vinylpyrrolidone) (PVP) are commonly used water-soluble polymers. These have been included in a wide variety of pharmaceutical products and are generally considered biologically inert [55,56], which makes them a good option for rapidly dissolving membranes or for increasing hydrophilicity in combination with an insoluble polymer. The high surface area of electrospun systems in comparison to traditional dosage forms means that acceptable doses can often be delivered without disintegrants or solubility enhancers. However, release from hydrophobic fibres is often bimodal, with an initial burst release caused by the dissolution of drug at the fibre surface, followed by slow release over the course of days or weeks limited by diffusion of drugs within the polymer [52]. This may not be a problem for inexpensive low toxicity drugs, where this partial release is sufficient to provide a therapeutic dose. Slow release is often desirable for other applications such as drug-eluting implants [57].

### 3.3. Bioadhesive Polymers

The moistness of the oral mucosa makes it a challenging site for adhesion; therefore, mucoadhesives are often required to achieve acceptable residence times. Depending on the nature of the polymer, several different mechanisms may be involved in mucoadhesion. These include the effects of surface tension, dehydration, diffusion, electrostatic interactions, and chemical adsorption (e.g., through the formation of covalent bonds). Water-soluble polymers such as PVP swell rapidly, causing the dehydration of the mucus layer [58]. The swelling results in intimate contact between the polymer and mucus glycoproteins and hydrates the polymers, further increasing the rate of diffusion into the substrate. Prolonged adhesion arises due to hydrogen bonding or ionic interactions, and entanglement, between the interpenetrating polymers and glycoproteins. In general, higher molecular weight water-soluble polymers result in improved residence times due to slower dissolution and increased chain length for interpenetration/chain entanglement [59]. More flexible polymers with linear-chain configurations tend to diffuse more easily into the biological tissue, resulting in an increased number of interactions and improved adhesion [60]. Highly coiled polymers, such as dextrans, are more bulky and less able to interpenetrate with the tissue. Polymers with similar surface chemistry to the glycoproteins are likely to be miscible and able to diffuse into the mucus [61]. Natural carbohydrates and protein polymers that have been widely reported to have mucoadhesive properties include chitosan [62], gelatin [63], hyaluronic acid [64], and alginates [65]. These polymers are charged at physiological pH; therefore, electrostatic interactions in combination with hydrogen bonding are likely involved in the adhesion mechanism. Synthetic polyionic polymers such as poly(acrylic acids), including Carbopol^®^ [66], and the Evonik Eudragit^®^ series have also shown mucoadhesive properties [34,67]. Thiolated polymers, such as thiolated chitosan [68] and thiolated hyaluronic acid [69], adhere by forming disulfide bridges with cysteine domains in mucins, resulting in adhesion through chemical adsorption.

### 3.4. Material Characterisation

No one method has been identified to measure mucoadhesion and there is no obvious correlation in results between methods, especially when compared to in vivo findings. The most common tests involve use of a texture analyser to measure mucoadhesive strength, the perpendicular force required to break, pull or peel away the sample from a model membrane [70,71,72]. Alternatively, in vitro residence time tests may be used to measure time until detachment, for example from ex vivo animal mucosa in a simulated saliva medium [34].

Different experimental in vitro set-ups have been used to quantify drug release kinetics. One such set-up is the paddle-over-disc method, which is specific for transdermal patches, measuring one-sided patch dissolution in a buffer at a paddle speed of 50–100 rpm. At the pre-determined time points, an aliquot is removed from the test solution and replaced with an equivalent volume of fresh buffer solution. The samples are then analysed by spectrophotometry techniques [73] or HPLC [74] to obtain a graph of the drug release over time. Simplified versions of this may also be performed by immersing the patch freely in the release medium [73] or adhered to a glass slide [75] and stirring the medium with a magnetic stirrer bar or laboratory shaker.

In vitro cell-based assays are important at the pre-clinical stage for evaluating irritation potential or side effects caused by the delivery system. At present, there are no internationally recognised standardised toxicity testes specifically for the oral mucosa. Cell metabolic assays are often used to give an indirect measure of cytotoxicity, for example the methylthiazoletetrazolium (MTT) assay that is recommended by the Organisation for Economic Co-operation and Development (OECD) to assess potential for skin irritation [76]. Although validated for tissue-engineered skin equivalents, these tests are also useful for oral mucosal studies due to the similarities in tissue structure. Similarly, the recently developed molecular test for skin irritation/sensitivity based on a validated gene signature profile (SENS-IS) [77,78] may also be of use in oral mucosal studies if this technology can be translated to oral tissue. Monolayers of cultured oral keratinocytes isolated from healthy tissue or keratinocyte cell lines can be used as in vitro models to test oromucosal materials [79,80]. Three-dimensional oral mucosal equivalents are increasingly preferred, being more physiologically representative and offering more accurate predictions of cell-toxicity while avoiding the use of animal models [81,82].

Drug diffusion across ex vivo oral mucosa may be measured using drug permeation chambers, such as the Franz diffusion cell. In such chambers, the tissue is placed between donor and receptor chambers, where the donor chamber contains the dose form and the receptor chamber holds a temperature controlled physiologically relevant buffer solution. In some instances, synthetic membranes may be used to eliminate some variability caused by animal tissue [83]. For these methods, the solution is removed from the receptor chamber at pre-determined time points to measure the drug permeation over time using a UV spectrophotometer or HPLC. More recently, the localisation of permeants within the oral mucosa itself has been visualised down to a micrometre scale resolution using matrix-assisted laser desorption ionisation-mass spectrometry imaging (MALDI-MSI) on sectioned tissue [84].

### 3.5. Excipients and Other Considerations

Where permeability is a limiting factor, permeation enhancers may be incorporated into the delivery system. There are a variety of classes of permeation enhancers with different modes of action that are outlined in a review by Sudhakar et al. [10]. In general, lipophilic uncharged drugs are more strongly affected by enhancers that increase membrane fluidity, such as fatty acids or laurocapram [85]. The mechanism of action of these classes is theorised to be a result of improved solubility of the drug in cell membranes, leading to faster uptake. Supporting evidence for this mechanism has recently been produced using permeation kinetic experiments and mass spectrometry imaging, showing co-localisation of the drug with the enhancer and increased capacity for the drug in the mucosa [86]. Hydrophilic drugs are generally more affected by surfactants, including bile salts, which are believed to extract lipids from the epithelium and form aqueous reverse-micelle channels within the tissue [85]. This increases the intracellular space available for paracellular transportation and thus increases the rate of permeation. At this stage, there has been little research involving electrospun systems containing permeation enhancers; however, many mucoadhesive polymers, in particular chitosan, can themselves enhance permeation by disrupting the structure of mucins and lipids at the mucosal surface [85].

In some cases, excipients have been included in electrospun systems to further enhance drug solubility, for example emulsifiers or complexing agents [87,88]. Nanoparticle drug delivery vectors, such as liposomes and polymersomes, have previously been incorporated into electrospun materials for a variety of non-oromucosal applications [89,90,91]. Research on oromucosal films containing nanoparticles has shown improvements to absorption and drug solubility and may protect certain drugs from enzymatic degradation [92]. However, due the additional manufacturing and regulatory complexity, these materials rarely make it to late stage clinical development.

Other considerations for a topical dosage form that adheres to the oral mucosa include disturbances to taste, speech, eating and drinking [15]. It may, therefore, be desirable to avoid foul-tasting drugs and excipients or deliver them unidirectionally into the mucosa. Low profile and flexible dosage forms may reduce the likelihood of the device becoming dislodged by mechanical forces in the mouth; therefore, flexible polymeric films or patches may be preferable to more traditional tablets in this regard.

## 4. Therapeutic Applications Currently in Development

### 4.1. Anti-Inflammatory

Chronic inflammatory diseases in the oral cavity are often mediated by dysregulated immune responses initiated by pathogens, foreign bodies, ionising radiation, or autoimmune disorders. Common chronic inflammatory diseases include oral lichen planus (OLP), which produces white lesions affecting 1–3% of the world’s population, and recurrent aphthous stomatitis (RAS), also known as aphthous ulcers or canker sores [93,94]. The aetiology of many chronic oral inflammatory diseases is poorly understood and no prophylactic treatments are available. Instead, corticosteroids or other anti-inflammatory agents are commonly used to manage the severity of ulcers and lesions. Systemic corticosteroid delivery often results in unacceptable side effects, whereas existing topical formulations, such as rinses, lozenges, and ointments, must be reapplied several times per day and result in inconsistent dosing. Topical corticosteroids are also associated with some serious adverse effects, including adrenal suppression and secondary candidiasis [95]; therefore, formulations that allow the specific delivery of well-defined doses are desirable. Ulcers and lesions are often highly sensitive; therefore, mucoadhesives, once carefully applied, may also prevent pain by providing a protective barrier against mechanical stimulation.

Several research groups, including us, have been at the forefront in the development of electrospun mucoadhesive patches (Table 2). Indeed, our close industrial collaboration with AFYX Therapeutics has enabled us to push this technology closer towards patient use. As part of our portfolio of studies, Colley et al. conducted a pre-clinical study on electrospun patches loaded with the corticosteroid clobetasol-17-propionate to treat chronic oral inflammatory diseases. The patches consisted of a drug-loaded (up to 20 μg per patch) layer of mucoadhesive fibres consisting of PVP and Eudragit^®^ RS100 with poly(ethylene oxide) (PEO) particles electrospun from 97% ethanol [34]. These polymers are all known to have mucoadhesive properties. Water-soluble PVP allows the patches to swell rapidly and insoluble RS100 allows the patches to maintain their structural integrity, improving residence time. A hydrophobic backing layer was introduced to promote unidirectional delivery and improve mechanical properties by electrospinning a second layer of poly (caprolactone) from 9:1 DCM/DMF and melting in an oven to produce a continuous film. Drug-free patches displayed buccal residence times of 96 ± 26 min in human volunteers with good patient acceptability (Figure 4). Drug-loaded patches released 80% of the drug over a 6 h period. In vitro cytotoxicity testing with tissue-engineered oral mucosal equivalents suggested that the patches were non-irritant [82]. This formulation is now a proprietary technology of AFYX Therapeutics with the brand name Rivelin^®^ and recently successfully met the primary end point in phase 2b clinical trials (ClinicalTrials.gov identifier: NCT03592342) for the treatment of OLP, showing a significant reduction in ulcer area, and is on track to become the first such electrospun mucoadhesive on the market.

Wei et al. used needleless electrospinning with a double ring-shaped spinneret for the rapid production of 3-layer composite meshes consisting of a layer of mucoadhesive PEO nanofibres electrospun from water containing 30% w/w diclofenac sodium, and a layer of hydrophobic poly (L-lactic acid) (PLLA) nanofibres electrospun from 1,1,1,3,3,3-hexafluoro-2-propanol (HFP) containing curcumin at up to 4% w/w (Figure 5). Curcumin was used as a model anti-inflammatory agent that may be beneficial for the treatment of RAS and diclofenac sodium, an antimicrobial analgesic, to reduce the risk of infection and relieve pain. The fibres were then placed onto a hypromellose-based gel in a mould and allowed to dry to produce an adhesive backing layer. Diclofenac sodium was shown to inhibit the growth of *Staphylococcus aureus* by placing the patches onto a bacterial lawn grown on a blood agar plate. Curcumin was shown to maintain its anti-inflammatory properties by measuring reduced pro-inflammatory gene expression by activated human monocytes. The release of curcumin from the fibres was sustained over a period of two weeks [41]. A relatively slow release, which is typical of hydrophobic fibres loaded with a hydrophobic drug and would potentially be disadvantageous for more expensive drugs, given that shorter residence times are more appropriate for RAS ulcers. The multiple layers of fibres make the system suitable for the co-administration of water-soluble and insoluble drugs, which is useful for inflammatory diseases, where a combination of different therapeutic agents may be beneficial (for example corticosteroids, antimicrobial agents, and analgesics).

Although there have been relatively few studies on electrospun mucoadhesives for inflammatory conditions, they have great therapeutic potential due to the large number of patients affected. Electrospun patches offer improved residence times over topical ointments and rinses and, unlike buccal tablets, are flexible and, therefore, less likely to place mechanical stress on sensitive lesions and ulcers. There is potential for further research in this area to develop formulations and for the co-administration of antifungal agents to counteract secondary oral candidiasis often observed with corticosteroid treatments.

### 4.2. Local Anaesthesia and Analgesics

Chronic oral mucosal pain is a common complaint that can have a wide variety of causes including infections, inflammation, chemotherapy, or surgery [100]. Over-the-counter oral non-steroidal anti-inflammatory agents (NSAIDs) and paracetamol are effective for oral pain management, but with some side effects associated with long-term use. Topical anaesthetics, such as lidocaine, are also highly effective for local pain relief and are commonly applied as gels or lozenges. NSAIDs can cause or delay the healing of oral ulcers, and so may not be appropriate for all kinds of oral pain [101]. Over 50% of patients undergoing treatment for head and neck cancer suffer from oral mucositis [102], a disruption in the oral epithelium, leading to painful inflammation and ulceration. Magic mouthwashes are a commonly used palliative treatment, typically containing combinations of local anaesthetics (lidocaine) or antihistamines, antimicrobial agents, corticosteroids, and coating agents. These have unclear effectiveness and often result in side effects [103]. Some studies suggest that morphine mouthwashes provide superior pain relief with reduced side effects [104,105]. Anaesthetic injections are used for some dental procedures; however, dental injections are the cause of dental anxiety for many patients [106]. Alcohol-based topical solutions may be applied using a cotton swab as an alternative. These have an unpleasant taste and can spread across the oral mucosa uncontrollably [107]. Electrospun mucoadhesive patches may offer another useful delivery method for dental anaesthesia or the treatment of chronic pain with improved site-specificity and prolonged delivery in comparison to rinses and ointments and with a lower profile and improved flexibility over adhesive tablets.

Rapidly dissolving electrospun membranes were previously developed as a potential delivery method for dental anaesthetic. Illangkoon et al. successfully fabricated electrospun PVP fibres loaded with mebeverine (up to 30% w/w), a drug with several applications including as a local dental anaesthetic. As would be expected of high surface area fibres of a water-soluble polymer, dissolution studies showed very rapid release, with the fibres dissolving within 10 s, allowing for a convenient application method with improved dissolution over the neat drug [73].

Clitherow et al. investigated the Rivelin^®^ formulation, consisting of drug-loaded fibres of blended PVP and Eudragit^®^ RS100 with a PCL backing film, for the delivery of lidocaine HCl to the oral mucosa for the management of prolonged pain and as a local anaesthetic. Lidocaine HCl was loaded into the fibres at 2.5% w/w. The patches released approximately 80% of the loaded lidocaine within 1 h and permeation experiments showed a permeability of 136 μg cm^−2^ min^−1^ in ex vivo porcine buccal tissue. Additionally, lidocaine released from the patches inhibited veratridine-mediated opening of voltage-gated sodium channels in SH-SY5Y neuroblastoma cells in a real-time functional assay, showing that therapeutic activity was maintained. The distribution of lidocaine in porcine buccal mucosa was imaged using MALDI-mass spectrometry to show time-dependent permeation, providing for the first time strong evidence of the electrospun patches’ efficacy as a local delivery method for dental anaesthetic to the oral mucosa (Figure 6) [93].

Oral pain represents a large market with multiple unmet clinical needs and is, therefore, a promising application for electrospun systems. Multiple studies have reported suitable release of lidocaine HCl from biocompatible mucoadhesive materials and some early results show effective targeted delivery. Further in vitro and in vivo investigation is expected in the near future. There is also scope to investigate the versatility of electrospun fibres for the delivery of alternative agents, which may be more effective for treating oral mucositis, such as benzydamine HCl, opiates, and amylmetacresol/dichlorobenzyl alcohol [103,108].

### 4.3. Antimicrobials

Oral candidiasis (OC) is caused by the opportunistic overgrowth of *Candida*, most commonly *Candida albicans* in the oral cavity. It is common in predisposed patients, such as people with dentures, diabetics, immunocompromised patients, and those on long-term antibiotic or steroidal therapy [109]. The infection can be present as superficial plaques, red lesions, or chronic plaques caused by fungal invasion of the epithelium. In some cases, OC may cause burning sensations, unpleasant tastes, or difficulty swallowing. Topical antifungal steroid rinses containing nystatin or miconazole are the first-line treatment and are usually effective. Even the most well tolerated antifungal rinses are sometimes associated with side effects including vomiting and diarrhoea and their high sucrose content can exacerbate other conditions such as tooth decay and diabetes [109]. Although rinses are effective when applied 4 times per day, there is potential to minimise side effects using a specific delivery method. Sustained release through mucoadhesives patches may allow the minimum inhibitory concentration to be maintained without requiring such a high initial dose, thus reducing side effects. Recent increases in antifungal resistance show a need for alternative antifungal therapies [110]. Some alternative therapies that have been explored include surfactants [111], synthetic peptides [112], and fatty acids [113]. Nanofibre encapsulation can enhance drug solubility and may be useful for the delivery of alternative antifungal agents that are incompatible with rinses.

Tonglairoum et al. developed electrospun PVP fibres with hydroxypropyl-β-cyclodextrin to rapidly release and improve the solubility of clotrimazole, a poorly soluble antifungal drug, for the treatment of OC. PVP was used as a rapidly dissolving polymer and the cyclodextrin as an excipient to form inclusion complexes to enhance drug solubility. Clotrimazole was loaded at up to 20% by dry mass and the fibre mats electrospun from mixtures of ethanol, water, and benzyl alcohol. The fibres rapidly dissolved in artificial saliva and were effective at eliminating the viability of *C. albicans* and *C. dubliniensis* suspensions within 2 h [87]. To prolong the effect, the material was further developed into a sandwich patch by electrospinning a second mucoadhesive layer from water consisting of 5:1 PVA/thiolated chitosan. The resulting sandwich patches released clotrimazole at a rate more suitable for prolonged antimicrobial effect, with approximately 70% released within 4 h [79].

Similarly, Szabó et al. incorporated terbinafine HCl at approximately 7% w/w into 1:5 chitosan/PVA fibres from an aqueous solution. The fibres dissolved rapidly in artificial saliva, releasing all of the drug within four minutes. In silico modelling with GastroPlus™ software predicted that up to 66% of the dose would be absorbed in the oral cavity if oral transit is properly regulated [88].

Aduba et al. also developed an electrospun material for the delivery of poorly soluble antifungal agents against oral candida. The 1:1 gelatin/nystatin fibres were electrospun from 1,1,1,3,3,3-hexafluoro-2-propanol and subsequently immersed in PEG diacrylate and 2,2-dimethoxy-2-phenylacetophenone as a photoinitiator dissolved in ethanol. Removing and curing using UV exposure produced cross-linked fibres with improved structural stability in aqueous solutions. The release rate was dependent on the degree of cross-linking and relatively slow, with approximately 20–70% released within 24 h. However, the authors did not assess the effectiveness of their system in any biological assays [94].

Clitherow et al. incorporated various unsaturated fatty acids as alternative antifungal agents into both the PCL and PVP/Eudragit^®^ RS100 of the Rivelin^®^ patch formulation at loadings of up to 22% and 12% w/w, respectively. Unlike in previous studies, disk diffusion inhibition and biofilm viability assays were used to demonstrate the potential of the patches at inhibiting both wild-type and azole-resistant *C. albicans* when applied directly to biofilms, thus clearly showing the effectiveness of mucoadhesive electrospun patches at treating OC. Dodecanoic acid was found to be the most effective of the fatty acids tested against pre-existing *C. albicans* biofilms [95].

Edmans et al. incorporated lysozyme, an antimicrobial enzyme, into the Rivelin^®^ formulation at a loading of 1% w/w by mixing the aqueous lysozyme-containing proportion of the electrospinning solvent into the polymer solution shortly before electrospinning. The patches released the enzyme at a suitable rate, with 90% of the enzyme released within 2 h. The enzyme was shown using an enzymatic assay to maintain high activity and inhibited the growth of the oral bacterium *Streptoccocus ratti* in suspension [96]. Lysozyme is effective against *C. albicans* and various oral Gram-positive bacteria and may be particularly useful as a treatment in patients with reduced saliva lysozyme concentration, such as children with chronic tonsillitis and patients with oral mucositis [114] or xerostomia (dry mouth). Perhaps more importantly, this work suggests that the formulation is suitable for delivery of biologics, which are particularly challenging to deliver with existing dose forms and have a variety of potential new applications in oral health, including as agents against bacterial, fungal, and viral infections [115].

Research so far has shown that a wide variety of alternative antifungal agents that could overwise not be delivered using rinses can be incorporated and released from electrospun mucoadhesives and one study has shown effectiveness against biofilms in vitro. It is expected that further in vitro and in vivo research will be performed to translate these materials for clinical use.

## 5. Future Research and Conclusions

A range of electrospun materials have been developed which incorporate and release drugs for treating oral diseases. These include rapidly dissolving membranes to allow easy administration of poorly soluble drugs through to devices that adhere for hours, delivering sustained doses. However, it remains challenging to evaluate and compare mucoadhesive and mechanical performance due to the lack of standardised mucoadhesion tests. Although several different electrospun devices for oral medicine are under development, the Rivelin^®^ patch is the only device so far that has been tested both in vitro and in humans to show a suitable residence time, of approximately 2 h, and good patient acceptability. Future drug delivery devices are likely to bring other advantages, such as longer residence times or the ability deliver drugs that are incompatible with non-aqueous electrospinning solvents. Therefore, it is expected that more electrospun drug delivery materials will be developed with an emphasis on clinical translation. So far, the technology has been applied to a narrow range of oral conditions as an improvement on exiting treatments; however, it is expected that, as the technology matures, it will enable more unmet clinical needs to be addressed.

Biologics are a class of therapeutics that are challenging to deliver using traditional dose forms and are currently almost exclusively delivered systemically using injections. Potential oral health applications include antimicrobial peptides and proteins to treat resistant infections or target specific strains of pathogen [116,117]. Certain cytokines, such as keratinocyte growth factor, show potential for regenerating the oral mucosa [118], for example following damage caused by oral mucositis, and could be effective if delivered directly to the affected site rather than systemically. Moreover, the delivery of therapeutic monoclonal antibodies directed at pro-inflammatory cytokines would radically alter treatment options for inflammatory disorders such as OLP or RAS, conditions where these molecules are a major driver of pathogenesis [119]. However, to deliver biologics, both the compatibility with the solvent system and the permeability of the mucosa must be considered. Indeed, recent work by Edmans et al. suggests that the Rivelin^®^ formulation is suitable for protein delivery [96]. Stie et al. recently reported mucoadhesive chitosan/PEO fibres electrospun from a mildly acidic aqueous solution, which are expected to be further investigated for oromucosal peptide delivery [120]. It is also expected that more complex electrospinning techniques for mixed or multi-domain fibres could be used to encapsulate biologics that would otherwise be susceptible to denaturation by the electrospinning solvent. Although biologics tend to permeate the oral mucosa less easily than small-molecule drugs, certain peptides, including insulin [121] and salmon calcitonin [122], have been observed to permeate the oral mucosa sufficiently to achieve therapeutic blood plasma concentrations. Therefore, it is expected that biologics could also permeate into the mucosa to provide a local therapeutic effect. Future work is needed to investigate permeation following release from a mucoadhesive electrospun material. Mucoadhesive patches may improve permeability by disrupting the superficial epithelium and providing intimate contact. Many conditions, including RAS, oral mucositis, and oral wounds result in an impaired epithelial barrier that may allow delivery to the target area. If necessary, further enhancement could be achieved using conventional permeation enhancers or drug delivery vectors [15].

Another potential use for electrospun mucoadhesive is to prevent and treat alveolar osteitis (dry socket), a painful condition caused by the lack of a blood clot at the site of tooth extraction [123]. The device could act as protective cover to prevent loss of the blood clot or protect underlying bone and nerves and deliver pain relief. This would likely require a device with a residence time of a few days.

Mucoadhesive patches could provide a local delivery method for new anti-tumour treatments for oral squamous cell carcinoma or pre-malignant lesions. Potential therapeutics include histone deacetylase (HDAC) inhibitors, such as suberoylanilide hydroxamic acid, which act as epigenetic chemosensitisers to increase the effectiveness of traditional chemotherapy [124,125]. Tyrosine kinase inhibitors, such as gefitinib or cetuximab, target epidermal growth factor receptors, which are overexpressed in many solid tumours, and can increase cancer cell apoptosis and radiosensitivity [126,127]. Imiquimod, an immune response modifier available as a dermal cream, has shown promise in animal models for reducing oral leukoplakia [128].

In conclusion, electrospun mucoadhesives make use of a scalable and industrially proven manufacturing process and are highly versatile in the range of drugs they can incorporate. They are attractive for drug delivery to the oral mucosa in that they are flexible and have a high surface area for drug release and, unlike existing dose forms, allow targeted delivery and prolonged retention times. It is envisioned that electrospun drug delivery devices will expand the range of treatments that can be applied to the oral mucosa and will have wide-ranging implications for the treatment of oral diseases.

## Figures and Tables

**Figure 1 pharmaceutics-12-00504-f001:**
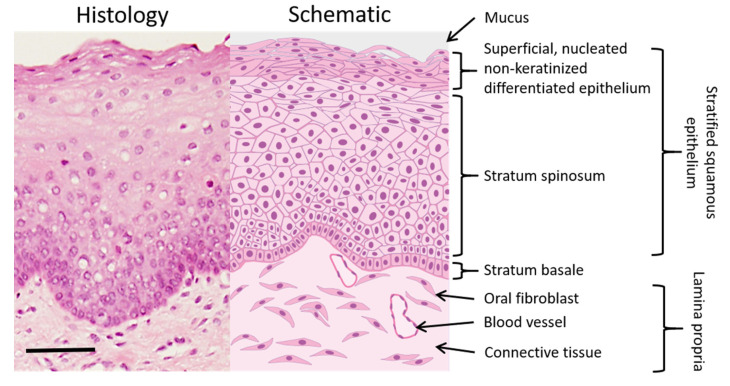
Histological (**left**) and schematic (**right**) image of the buccal oral mucosa (histological image courtesy of Prof. Keith Hunter, Unit of Oral Pathology, University of Sheffield). Scale bar = 100 μm.

**Figure 2 pharmaceutics-12-00504-f002:**
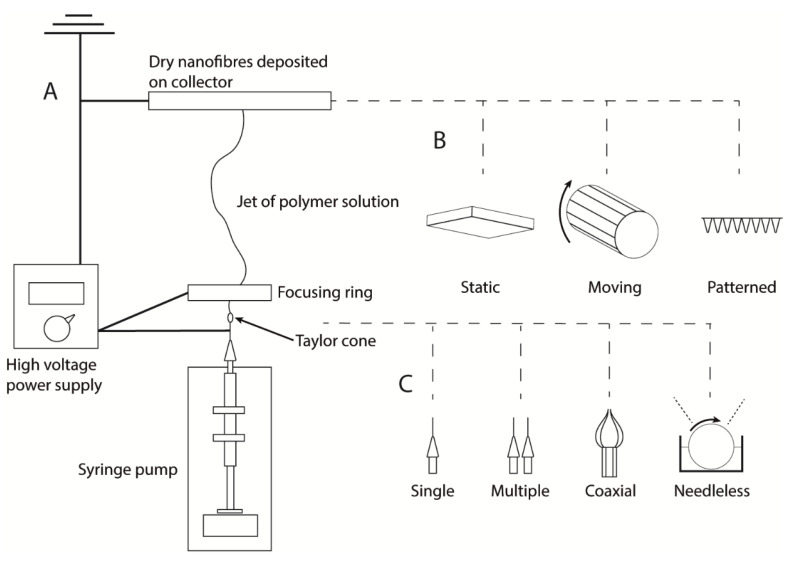
(**A**) Schematic diagram of typical electrospinning apparatus. A high voltage power supply injects charge into the metallic syringe tip, causing a polymer jet to be ejected towards the grounded collector plate. The jet dries during flight, depositing a nanofibre mesh. (**B**) Static collectors result in a random mesh of fibres; moving collectors can be used to produce aligned fibres; patterned collectors result in textured membranes. (**C**) Multiple needles in combination with a moving collector allow increased output or the production of mixed fibre types; coaxial needles allow the production of core-sheath fibres with multiple polymer domains; needleless spinnerets allow many polymer jets to be produced simultaneous to give increased output.

**Figure 3 pharmaceutics-12-00504-f003:**
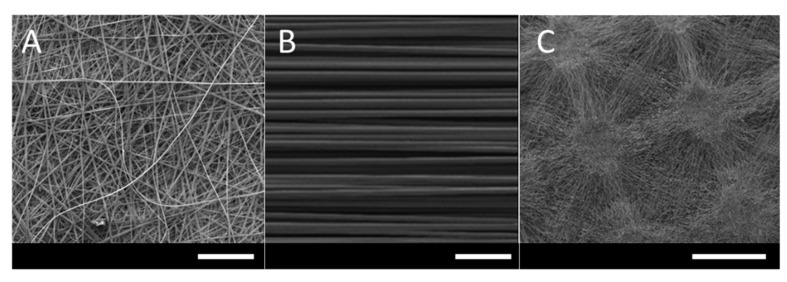
(**A**) Scanning electron microscopy image of fibres electrospun from a solution of Eudragit^®^ RS100 and poly(vinylpyrrolidone) in ethanol/water using a static collector. Scale bar = 100 μm. (**B**) Aligned polyhydroxyalkanoate blend fibres electrospun from chloroform using a rotating cylinder collector. Scale bar = 100 μm. Reproduced from [45], John Wiley & Sons Ltd., 2019. (**C**) Poly (3-hydroxybutyrate-co-3-hydroxyvalerate) fibrous membranes with rectangular micropatterns electrospun from dichloromethane/methanol using a micropatterned static collector. Image courtesy of Dr Ílida Ortega Asencio, University of Sheffield. Scale bar = 1 mm.

**Figure 4 pharmaceutics-12-00504-f004:**
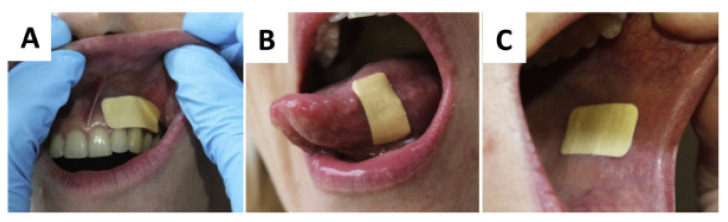
Mucoadhesive Rivelin^®^ patches placed on the (**A**) gingiva, (**B**) lateral tongue, (**C**) buccal mucosa of a healthy human volunteer. Reproduced from [82], Elsevier Ltd., 2018.

**Figure 5 pharmaceutics-12-00504-f005:**
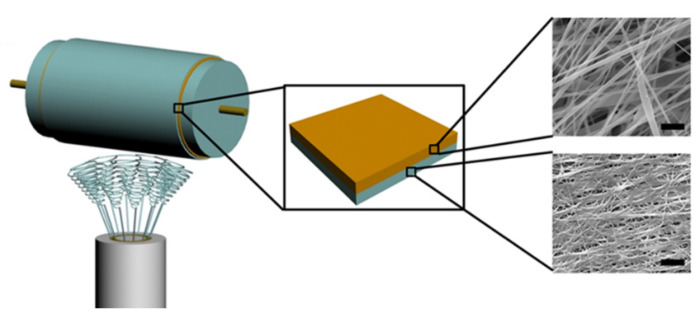
Fabrication of multi-drug-loaded bilayer composite meshes using double-ring slit needleless spinneret. Yellow layer: curcumin-loaded PLLA nanofibre mesh; blue layer: diclofenac sodium-loaded PEO nanofibre mesh. Reprinted with permission from [41]. Copyright (2019) American Chemical Society.

**Figure 6 pharmaceutics-12-00504-f006:**
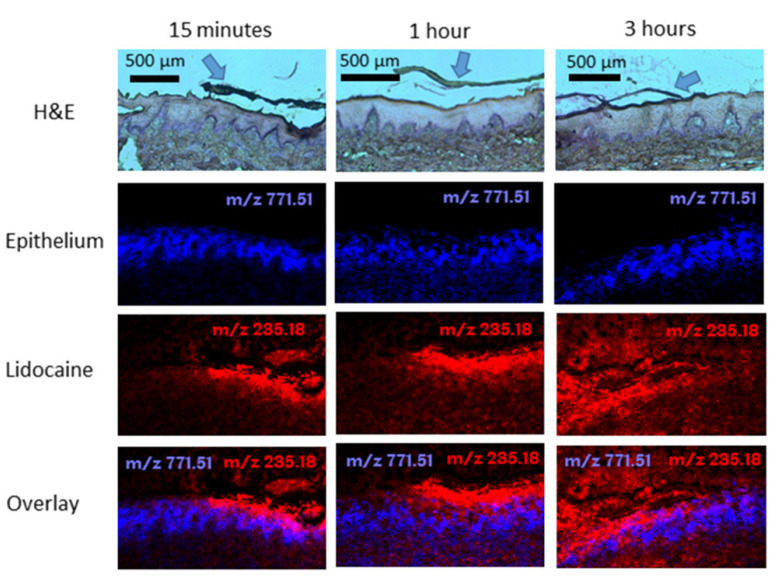
Haematoxylin and eosin (H&E)-stained tissue sections and corresponding MALDI-MS images of porcine buccal mucosa exposed to dual-layer electrospun patches containing 3% (w/v) lidocaine HCl (m/z 235.1805 [M + Na]+; red) after 15 min, 1, and 3 h. The epithelium (blue) for each sample is shown using the epithelial marker lipid phosphatidylglycerol (34:1) (m/z 771.5140 [M + Na] +). The arrows in the H&E images show the position of the electrospun patch. Reproduced from [96]. Copyright (2019) American Chemical Society.

**Table 1 pharmaceutics-12-00504-t001:** Variations and modifications to the electrospinning technique.

Modification	Type	Application	Reference
Focusing ring	Attachment	Improved performance	[44]
Rotating mandrel collector	Collector	Aligned fibres, more uniform membrane thickness	[45]
Belt collector	Collector	Increased output	[46]
Patterned collector	Collector	Patterned membranes	[37]
Coaxial	Spinneret	Multiple domain fibres	[47]
Side-by-side	Spinneret	Multiple domain fibres	[48]
Simultaneous electrospinning	Spinneret and collector	Mixed-fibre membranes	[49]
Multi-needle	Spinneret	Increased output	[49]
Needleless	Spinneret	Increased output	[42]
Centrifugal	Spinneret	Increased output	[43]
Emulsion	Feedstock	Multiple domain fibres	[39]
Sequential electrospinning	Feedstock	Multiple-layered membranes	[49]
Melt	Feedstock	Solvent-free manufacture	[50]
In Situ mixing	Feedstock	Porous fibres	[51]

**Table 2 pharmaceutics-12-00504-t002:** Electrospun mucoadhesives under development for use in oral health.

Indication	Polymer	Drug	Solvent	Processing	Ref.
Oral lichen planus	Adhesive/drug release: PVP, Eudragit^®^ RS100 Backing layer: PCL	Clobetasol-17-propionate	97:3 ethanol/water 9:1 DCM/DMF	Sequential electrospinning, heat treatment	[82]
Recurrent aphthous stomatitis	Lower layer: PEO Upper layer: PLLA	Diclofenac sodium Curcumin	Water HFP	Double-ring slit needleless spinneret	[41]
Pain relief	Adhesive/drug release: PVP, Eudragit^®^ RS100 Backing layer: PCL	Lidocaine	97:3 ethanol/water 9:1 DCM/DMF	Multiple-layer electrospinning, heat treatment	[96]
Oral candidiasis	PVP	Clotrimazole Excipient: hydroxypropyl-β-cyclodextrin	7:2:1 ethanol/water/ benzyl alcohol	Conventional electrospinning	[87]
	PVPBacking layer: PVA/thiolated chitosan	Clotrimazole Excipient: hydroxypropyl-β-cyclodextrin	7:2:1 ethanol/water/ benzyl alcohol	Sequential electrospinning	[79]
	PVA/chitosan	Terbinafine hydrochloride	Water	Conventional electrospinning	[88]
	gelatin	Nystatin	HFP	Electrospinning, UV cross-linking	[97]
	Adhesive/drug release: PVP, Eudragit^®^ RS100 Backing layer: PCL	Dodecanoic acid	97:3 ethanol/water 9:1 DCM/DMF	Sequential electrospinning, heat treatment	[98]
Antibacterial	Adhesive/drug release: PVP, Eudragit^®^ RS100 Backing layer: PCL	Lysozyme	97:3 ethanol/water 9:1 DCM/DMF	Sequential electrospinning, heat treatment	[99]

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
