# Peer review of "Mucoadhesive Electrospun Fibre-Based Technologies for Oral Medicine"

_pharmaceutics, 2020, doi:10.3390/pharmaceutics12060504_

Round 1

Reviewer 1 Report

The current review article provide very good overview on electrospum technology for oral mucosal drug delivery and it can be accepted in current format.

Reviewer 2 Report

The review on oromucosal delivery by Edmans et al. addresses the potential of electrospinning in terms of a range of small molecules cargoes match to disease areas localised to the mouth.  It is a well-written analysis by a team who are clearly experts in the area.  The Figs are very nice and they are mostly original.  The basic physiology is well presented alng with the principles behind the electrospinning technology, and an up to date table on products.  It is excellent.

A few very minor quibbles that could be easily addressed:

  • Readers unfamiliar with the team might not know the link from the authors to the Rivelin technology, so this should be declared up front and not just in the declaration at the end of the article.  A single line at the beginning of its discussion would do.
  • The authors should ensure balance with other technologies apart from Rivelin. A line placing it in context with competition would be helpful.
  • Fig 1 is very nice.  Please include a vertical scale bar.
  • Line 126/127:  What does "low profile" mean?  Suggest deleting that phrase.
  • line 248:  authors advocate MTT tests for irritation according to OECD recommendations. Surely, other validated in vitro tests related to inflammatory parameters need to be added as MTT only measures mitochondrial parameters and proliferation, not irritancy directly ?  Seems very limited.
  • line 256:  The Ussing chamber cannot be used for dosage forms, just solution-solution flux. Franz Cells are the main option here.
  • line 504:  The authors suggest that Rivelin can be used for protein delivery. A line to support this would help as this is entirely different from small milecule permeation. 

Reviewer 3 Report

I am pleased to recommend publication of this Review articles in Pharmaceutics. It is well-written in a clear way. The future research and conclusion is informative and inspiring. I have only one small concern and am interested to know more.

(1) surface chemistry should be an  important factor affecting the mucoadhesion. In this manuscript, electrospun fibre-based polymers have improved mucoadhesion ability due to the large surface area. this is a very important property and is very well described and summarised. could the authors also provide a bit more information about effects of surface chemistry of electrospun fibre on mucoadhesion? I can see one paragraph about "Bioadhesive polymers", may be the author can enrich this paragraph, either by some more text writing or some schematic. For example, what is "chemical adsorption"? How dose the "interpenetration" look like? how do the polymer interpenetrate with oral tissue?
